# Processing Speed throughout Primary School Education: Evidence from a Cross-Country Longitudinal Study

**DOI:** 10.3390/bs13100873

**Published:** 2023-10-23

**Authors:** Tatiana Tikhomirova, Yulia Kuzmina, Artem Malykh, Sergey Malykh

**Affiliations:** 1Department of Psychology, Lomonosov Moscow State University, 125009 Moscow, Russia; tikho@mail.ru; 2Psychological Institute of Russian Academy of Education, 125009 Moscow, Russia; papushka7@gmail.com (Y.K.); malykhartem86@gmail.com (A.M.)

**Keywords:** processing speed, developmental trajectories, primary school education, cross-country longitudinal study, mixed effects growth modeling, latent class growth modeling

## Abstract

This cross-country four-year longitudinal study investigated the development of processing speed throughout primary school education. The analyses were conducted on data accumulated from 441 pupils in grades from 1 to 4 (aged 6.42 to 11.85 years) in Kyrgyzstan and Russia. Mixed effects growth modeling was applied to estimate average and individual growth trajectories for processing speed in two cross-country samples. Latent class growth modeling was conducted to describe various types of growth trajectories for processing speed and to compare the distribution of the types within the analyzed samples. According to the results, processing speed significantly increases across primary school years. The trajectory is described by nonlinear changes with most dynamic growth between grades 1 and 2, which slows down until grade 4. No significant cross-country differences were found in the initial score of processing speed or developmental changes in processing speed across primary school years. The development of processing speed is described by a model including three quadratic growth types but this minimally differs. It is concluded that in both samples, the development of processing speed may be characterized by homogeneity, with the most intensive growth from grade 1 to grade 2 and subsequent linear improvement until grade 4.

## 1. Introduction

Processing speed is a basic cognitive ability responsible for the accuracy and speed of processing information and underlies individual differences in higher-order cognitive abilities (e.g., intelligence), educational achievement, and well-being [1,2,3,4,5]. Moreover, information-processing speed may be a rehabilitation target for the prevention measures of psychological stress, especially for individuals with atypical development and those living in a disadvantaged SES [6]. It has also been shown that individual differences in the developmental trajectories of processing speed affect the efficiency of functioning of the entire human cognitive sphere throughout life [3,7,8,9,10]. The “dramatic decrease” in information-processing speed with age and the high correlation with general cognitive abilities, as shown in previous studies, makes it possible to consider this cognitive indicator (measured using choice reaction time) a potential biomarker of cognitive age and, possibly, biological age in general [11] (p. 19).

Longitudinal studies involving multiple measurements of the cognitive trait in the same respondents reveal age-related changes in processing speed throughout life [12,13,14,15]. A sharp increase was shown throughout childhood, followed by a plateau in adolescence, reaching asymptotic values, and then a gradual decrease in processing speed throughout adulthood [13,14]. The importance of the method of measurement of processing speed was especially emphasized [12]. Studies have shown that the development of processing speed is better described by exponential and quadratic models than linear, hyperbolic, inverse regression, and transition models [12,16,17]. At the same time, despite the observed nonlinear pattern of development, there is a possibility that latent classes of development of processing speed exist that have escaped the attention of researchers. Thus, even with a relatively homogeneous sample in the context of a growth pattern, latent groups of respondents may exist that differ not only in the initial levels of development of a cognitive trait and the pace of its change but even in the direction of change [18,19]. For example, for approximate number sense, a cognitive trait responsible for the ability to estimate and compare quantities without the use of symbols, a general nonlinear pattern of changes was observed with two latent classes: the first class was characterized by growth, whereas the second showed no significant growth and included 36% of individuals [20].

Indeed, the pattern of change in developmental trajectories that is most suitable for describing group data may not be as useful regarding characterizing individual trajectories of the development of processing speed in individual respondents during a certain age period [21]. Primary school age can be considered the most sensitive period regarding the study of changes in the development of information-processing speed—the period when, according to neurophysiological studies, the myelination process, which is involved in the formation of various aspects of attention, is most intensive [22,23]. Additionally, the process of learning in school begins during this age period, which actualizes the development of processing speed in the conditions of active assimilation of new knowledge, rules, and concepts.

Only a handful of longitudinal studies have been conducted on the development of information-processing speed throughout primary school education. In one such study, which analyzed the development of processing speed in childhood and adolescence, it was found that the developmental trajectory is best described by a quadratic function, demonstrating a sharp increase in processing speed from the age of 5 to 18 years [24]. However, this study was limited in its design, and some respondents were tested only twice with an interval of two years, while some were tested four times with an interval of six months [24]. Therefore, the trajectory of processing-speed development was calculated from measurements performed on different respondents using two different tests, which limits the understanding of the periods of the most intensive development during childhood and adolescence. The nonlinear nature of the changes from age 7 to 12 years was confirmed in a longitudinal study that included typically developing children and their peers with attention deficit hyperactivity disorder [22]. In particular, it was shown that the development of processing speed, measured by the choice reaction time, is characterized by a nonlinear pattern of change. At the same time, in both analyzed samples of children with typical and atypical development, the processing speed increased most intensively from 7 to 9 years [22]. Similar results were obtained in a longitudinal reaction time study using tasks with four choices [17]. These results indicated a nonlinear trajectory of development during the period of primary schooling, with the most intensive decrease in reaction time from the first to the second year of education (from age 7.8 to 8.9 years) and then a stable decrease until the fourth year (up to age 10.8 years) [17].

In most longitudinal studies, the emphasis is on the study of average (for the analyzed sample) trajectories of the development of processing speed, whereas individual trajectories are omitted. Meanwhile, simultaneous analyses of average and individual developmental trajectories open up the possibility of understanding whether the rates of change in processing speed differ between study participants or whether these changes are related to the group as a whole. It becomes possible to show whether processing speed differs between the study participants at each moment of measurement or, on the contrary, it differs to a greater extent for one participant over a certain time interval. Importantly, the analysis of individual trajectories opens up the possibility of identifying latent classes in the development of processing speed, which in turn allows for the identification of homogeneity or heterogeneity of the sample during the period of primary school education and helps to provide preventive assistance in the organization of school education.

Among basic cognitive abilities, processing speed is influenced by cultural factors to a lesser degree than, for example, intelligence, but is still a subject to them, primarily those related to formal education [25]. In particular, cross-cultural features in the change in the variability of the processing speed during the period of school education have been reported. It has been shown that the degree of narrowing the range of variability is directly proportional to the quality of school education in the country [26]. Among the educational factors, the foremost are the formal and content conditions of education in classes, schools, regions and countries: the number of students in one class, the organization of work and teacher qualifications, training programs, the language of instruction, the number of lessons in a particular subject, and the requirements of state educational standards [18,20,27].

Cross-cultural longitudinal studies of processing speed are few and far between, and their results are sometimes contradictory. For example, in one cross-cultural longitudinal study of the development of processing speed, a higher rate of development of this cognitive indicator was recorded among East Asian children from 4.5 to 11 years of age compared with their peers from the United States when the starting level was controlled at 4.5 years [27]. However, the same study reported that not all samples of East Asian children differed significantly from all samples from the United States regarding the dynamics of information-processing speed [27]. Several papers cover the age-related specificity of data on the development of information-processing speed obtained from different cross-cultural samples (e.g., [28]). For example, it was shown that only at older ages did Taiwanese respondents appear to be slower than their United States counterparts. However, these findings are based on an analysis of collective data from standardized Wechsler scales from the past two decades [28].

A promising approach to cross-cultural analysis of the trajectories of the development of cognitive traits during schooling involves samples of respondents from countries, on the one hand, with a similar organization of the education system, and on the other hand, with differences in the quality of schooling. As former republics of one union state, Kyrgyzstan and Russia share the same educational system, including four-year primary school education starting at 6.5 years of age, one primary school teacher for all lessons and years, and the same educational standards (for more details see [18]). However, the effectiveness of these countries’ two national educational systems differs, as demonstrated by their different SES ratings in the Human Development Report 2020 (https://hdr.undp.org/content/human-development-report-2020) (accessed on 10 August 2023). In particular, due to severe lack of teachers in Kyrgyzstan, a single primary school teacher works both morning and evening shifts, and the shortage of schools leads to a larger class size of 40 to 45 pupils in comparison with the 19–27 schoolchildren in Russian classes [20]. This difference enables the analysis of the developmental trajectories of processing speed and its types to be conducted in educational conditions that are similar in their formal organization but different regarding the quality of learning at primary school.

This cross-country four-year longitudinal study has the following main goals:To estimate both average and individual growth trajectories for processing speed across all period of primary school education;To evaluate country differences in growth trajectories of processing speed;To describe various types of growth trajectories for processing speed and to compare the distribution of the types for the Russian and Kyrgyz samples of primary schoolchildren.

For a number of years, we have been working on a “Cross-cultural Longitudinal Analysis of Student Success” (“CLASS”) project [20]. This longitudinal project explores various psychological characteristics—information-processing speed, visuospatial working memory, approximate number sense and number-line accuracy, intelligence, personality traits, intrinsic motivation, academic achievement, etc. Some results of the “CLASS” project have been published [17,18,20]. 

In order to achieve the goals of this study, longitudinal data for four years of the same group of schoolchildren—Russian and Kyrgyz participants in the “CLASS” project—were used. The Section 2 describes the sample of this study, which completely coincides with the sample of some previous studies, since these articles analyzed longitudinal data for four years of the same group of schoolchildren—Russian and Kyrgyz participants in the “CLASS” project. Our statistical approach describes a fairly standard approach to the statistical processing of longitudinal data on processing speed, which is identical to the approach applied to the processing of data on another cognitive traits. To select the models that best describe the data, identical criteria were applied as those which have been listed in previous articles [18,20]. 

Applying a unified approach to the analysis of the different cognitive traits will provide an opportunity to expand the understanding of scientific and educational communities about the peculiarities of the cognitive development across the primary school years.

## 2. Materials and Methods

### 2.1. Participants

Longitudinal data over a one-year interval of 441 pupils in grades from 1 to 4 in Kyrgyzstan and Russia who participated in the “CLASS” project were used.

The Kyrgyz sample included 303 participants (46% were girls). The mean age in the first grade was 7.46 years (SD = 0.39, range 6.42–8.83) and that in the fourth was 10.40 years (SD = 0.39, range 9.33–11.83). Sixty-six percent of the participants were Kyrgyz, 10% were Russian, and the rest belonged to other ethnic groups (e.g., Dungan, Uyghur, Kazakh). Some participants took part only once; thus, their data were removed from the analysis. Fifty-nine percent of the pupils participated in four waves, 31% in three waves, and 10% in two waves.

The Russian sample included 138 participants (46% were girls) from one primary school. The mean age in the first grade was 7.84 years (SD = 0.34, range 7.06–8.37) and that in the fourth was 10.77 years (SD = 0.36, range 9.72–11.85). One hundred percent of the participants were Russian. Some schoolchildren participated only once; thus, their data were removed from the analysis. Forty-eight percent of the pupils participated in four waves, 38% in three waves, and 14% in two waves.

### 2.2. Procedure

All pupils present on the testing day at school participated in the study. Testing of the participants took place in their schools at the end of each year throughout the primary school years. The participation of pupils in this research project followed strict adherence to the same protocol and instructions used in both samples. The instructions were given in Russian in both Russian and Kyrgyz samples. Consent from parents for the participation of their children and from participants and school authorities was obtained prior to testing. The analysis was performed on the deidentified data of the participants.

The Ethics Committee of the Psychological Institute of the Russian Academy of Education approved the “CLASS” project (protocol No. 2016/2–12). 

### 2.3. Measure

Processing speed was evaluated using the “choice reaction time” computerized test with four choices [17]. A detailed description of the “choice reaction time” test is provided in previous studies [17]. The accuracy and reaction time for correct responses in seconds were recorded. Thus, a lower reaction-time value corresponds to a higher processing speed.

### 2.4. Statistical Approach

In the first step, descriptive statistics and independent sample t-tests were applied to compare processing speeds for Russian and Kyrgyz samples in each grade across primary school education.

In the second step, mixed effects growth modeling (also known as multilevel regression) was applied to identify the average growth trajectory for processing speed and to estimate whether there were significant between-individual differences of changes in processing speed. Within the multilevel regression framework, it also becomes possible to assess whether the average development trajectory in processing speed corresponds to a linear or, conversely, nonlinear pattern. In addition, it becomes possible to evaluate how the rate of change in the processing speed is associated with variables that change over time and with variables that do not change over time, such as country of residence.

The reaction time for correct responses in the “choice reaction time” test was considered a dependent variable. The reaction time value was converted into Z-scores.

Several multilevel models were estimated. A more detailed description is presented in previous studies [18,20]. 

Baseline model (intercept-only). This model is recommended as a starting point for multilevel modelling. The model evaluates the mean predicted score at each point in time, interindividual variance (differences between individuals in terms of the trait at each point in time), and intraindividual variance (stability or instability of the trait over time for a particular individual). 

Model 1: Linear pattern. To estimate the time changes in processing speed, the variable “Time” is added. The coefficient of variable “Time” indicates significance, direction (positive or negative), and value of changes across time. A linear pattern implies that time changes are equal between each adjacent time points. This model uses a random intercept and fixed slope and random intercept. 

Model 2: Nonlinear pattern. A time-squared variable was added to identify which pattern of changes in processing speed fits the data better, linear or non-linear.

Model 3: Between-individual differences in the rate of change. To estimate the significance of between-individual differences in the rate of change of processing speed, random slopes of the variables “Time” and “Time-squared” were tested. This model estimates the variance of the slope(s) variables and covariance between the slope(s) and the intercept. 

Model 4: Cross-country differences in processing speed. A country variable (0 = Kyrgyz sample, 1 = Russian sample) was added to estimate differences in between Russian and Kyrgyz samples in the level of processing speed in each time point.

Model 5: Cross-country differences in the rate of change. To estimate between–country difference in the rate of change, interaction between the time variable and country variable was added. The significance and value of interaction term indicates the significance and value of between-country differences in the rate of change in processing speed.

In the third step, latent class growth modeling was applied to identify different types of growth trajectories of processing speed (latent classes). This step is similar to mixed-effect growth modelling, as it allows us to estimate the average growth trajectory and to identify which pattern of changes fits the data better. Then, models with a different number of latent classes (2, 3, 4) were tested. For each model with certain number of latent classes, we have also estimated models with different parameters of variance in latent intercept, linear change factor, and latent quadratic change factor. First, the variances was constrained to equal 0 in each latent class. Secondly, variances were freely estimated for each factor, but they were constrained as equal across latent classes. Thirdly, variances were estimated as free across latent classes. 

The selection of the most appropriate model was guided by several fit statistics (in particular, Bayesian information criterion (BIC), entropy, and Vuong–Lo–Mendell–Rubin likelihood ratio test and adjusted Lo–Mendel–Rubin LR test (VLMR LR and adjusted LMR LR, respectively). After a model with a certain number of latent classes was selected, we estimated cross-country differences in the proportion of each latent class.

Analysis was conducted with Stata 15.0 and Mplus 7.0 software.

## 3. Results

This study analyzed the processing speed in schoolchildren across all primary school years in two cross-country samples—Russian and Kyrgyz.

### 3.1. Descriptive Statistics

Table 1 shows the average values of reaction time to correct answers for the “Choice reaction time”, standard deviations, minimums and maximums at each grade during primary school education (in seconds). A lower average response-time value corresponds to a higher processing speed.

According to Table 1, the reaction time decreased over time in both samples, which implies that processing speed increased from grade 1 to grade 4. The variability in processing speed scores decreased in both analyzed samples from grade 1 to grade 4. Interestingly, although the size of the Russian sample was smaller than the size of the sample from Kyrgyzstan, the variance in reaction time was larger in the Russian sample.

### 3.2. Independent Samples t-Test

Table 2 shows the results of the comparison of processing speed between Russian and Kyrgyz samples in each grade across primary school education.

Table 2 shows that there were no significant differences in processing speed between the two samples in all grades except the second. In the second grade, participants from Russia were faster than participants from Kyrgyzstan.

### 3.3. Mixed Effects Growth Modeling

Within the mixed effects growth analysis, the four several multilevel models were first tested on the pooled sample: a baseline model (Table 3), a linear pattern growth model (Model 1 in Table 3), a nonlinear pattern growth model (Model 2 in Table 3), and a model with a random slope of the variable “Time” (Model 3 in Table 3).

Mixed effects growth modeling revealed that reaction time significantly decreased from the first to fourth grades, indicating an increase in processing speed. The coefficient of the time-squared variable (see Table 3, Model 2) was significant and positive; thus, the reduction in reaction time over grades slowed down.

The random slope model (Model 3) fitted the data significantly better than the model with a fixed slope, indicating that there exist significant between-individual differences in the growth of processing speed.

Furthermore, a country variable (0 = Kyrgyz sample, 1 = Russian sample) was added to the model (Model 4 in Table 4) and the interaction between the time and country variables was tested (Model 5 in Table 4).

The results of models including the country variable and with interactions with the time and time-squared variables (see Table 4) demonstrated that there were no significant differences between the Russian and Kyrgyz samples in initial level of processing speed and in the rate changes in processing speed.

The predicted average trajectories of processing-speed development across primary school education for two samples are shown in Figure 1.

Individual trajectories of development processing speed were estimated for the two analyzed samples from Model 3 (see Figure 2).

Thus, mixed effects growth modeling demonstrated that processing speed significantly increased during all periods of primary school education. According to the results, the trajectory of the processing speed is characterized by nonlinear changes. In particular, more intensive growth was observed between grades 1 and 2, which later slowed down until grade 4. The testing of models with the country variable and its interactions with the time and time-squared variables reveals that there were no significant cross-country differences in the initial level of processing speed or developmental changes in processing speed in the primary school years. The testing of models with a random slope of the time variable showed that the coefficient of the variable “Time” significantly varied across individuals, indicating that individuals significantly differed in changes in processing speed. This result may indicate that different types of growth trajectories of processing speed across primary school education exist.

### 3.4. Latent Classes Growth Modelling 

Within the latent classes growth analysis, a single-class growth model was first specified. The growth model with linear and nonlinear changes was estimated on the pooled cross-country sample. The fit indices are presented in Table 5.

The fit indices demonstrated that the model with quadratic changes had a lower sample-adjusted BIC, higher CFI, and lower SRMR (see Table 5). The root mean square error of approximation (RMSEA) was lower in the linear model, but the two models’ 90% CIs of RMSEA overlapped. Therefore, the quadratic model was selected as a better-fitting model.

Then, several multilevel models including 2, 3, and 4 latent classes were estimated. Models with different latent classes and different variances of the intercept, slope, and quadratic term were also estimated. The fit indices for each tested model are shown in Table 6.

Fit indices for models of growth of processing speed, presented in Table 6, demonstrated that the model with three latent classes with equal variances of I, S, and Q across classes had better fit indices. The VLMR and LMR LR tests confirmed that the model with three latent classes fitted significantly better than the model with two latent classes, while the model with four latent classes did not fit better than the model with three latent classes. The parameters of each latent class are shown in Table 7.

The results presented in Table 7 demonstrated that the third latent class is the most prevalent (92%). This class is characterized by a linear decrease in reaction time, indicating a steady improvement in processing speed from grade 1 to grade 4. The second latent class included 7% of the participants and was characterized by a fast growth in processing speed from grade 1 to grade 2 that subsequently slowed down significantly. The least numerous first latent class included only 1% of the participants, who demonstrated a medium growth of processing speed from grade 1 to grade 2 and a further decrease in processing speed (increase in reaction time) afterward. Since this class consisted of only a few cases, it can be considered an outlier.

The aforementioned results are illustrated in Figure 3, where the first class is represented by a red line, the second class is represented by a blue line, and the third class is represented by a green line.

A cross-country difference analysis of the proportion of latent classes was performed. According to the results, there were no significant differences between the analyzed countries in the proportion of latent classes (χ^2^(2) = 4.19, *p* = 0.12). The proportions of the three latent classes of the proceeding speed-growth trajectories within the Russian and Kyrgyz samples are shown in Table 8.

Thus, latent class growth modeling demonstrated that the development of processing speed is described by a model with three quadratic growth types: fast growth from grade 1 to grade 2 and then a consistent linear improvement (92%), fast growth from grade 1 to grade 2 and then a slowdown of growth (7%), and medium growth from grade 1 to grade 2 and then a lack of growth (1%). There were no significant differences between the analyzed countries in the proportion of developmental types. Although three latent classes can be statistically identified, the differences between the classes are minimal. Therefore, the sample is sufficiently homogenous regarding the development of processing speed across primary school education. Thus, the development may be characterized as intensive from grade 1 to grade 2 with a subsequent linear improvement until grade 4.

## 4. Discussion

In this study, we analyzed the development of information-processing speed during the whole period of primary school education using samples of schoolchildren from two countries: Kyrgyzstan and Russia. These countries have similarly organized educational systems but differ in socioeconomic status, which leads, among other things, to differences in the quality of schooling. For the first time in a cross-cultural longitudinal study, both the average trajectory of the development of information-processing speed and individual trajectories for Kyrgyz and Russian samples across primary school education were determined. The mixed effects growth modeling shows that over the course of all four years of primary education at school, information-processing speed increases in both samples of schoolchildren.

Additionally, the average trajectory of the development of information-processing speed is nonlinear. Thus, the most intensive growth occurred from the first to the second year of this study, and then, up to the fourth year of study, the growth slowed down somewhat but remained significant. In particular, at the start of the schooling period, the average reaction time was 1.29 and 1.26 s in the Russian and Kyrgyz samples, respectively, and by the second year of schooling, the reaction time had decreased to 1.06 and 1.14 s, respectively, indicating an increase in information-processing speed. A further reduction in reaction time from the second to the third and fourth years is less pronounced (see Table 1). These results are fully consistent with data from studies of the reaction-time trajectory obtained from a sample of Spanish and Russian children of primary school age, indirectly confirming the cultural universality of the development of this cognitive indicator [17,22]. In particular, the most intensive decrease in reaction time and, accordingly, increase in information-processing speed are reported in the age ranges of 7–9 years [22] and 7.8–8.9 years [17]. The data obtained in this study show that the intensive development of speed characteristics between seven and nine years of age is consistent with the data of neurophysiological studies on the increase in the length and diameter of axons and the continuation of the myelination process, which increases the speed and efficiency of nerve impulse conduction (e.g., [23]).

No cross-cultural differences were found either at the initial level of information-processing speed (during the first year of schooling) or in the direction or pace of its development throughout primary school education. Moreover, in the fourth year of study, the information-processing speed in both analyzed samples completely coincides and reaches a value of 0.95 s. Significant differences in information-processing speed between the Russian and Kyrgyz samples in the second year of study obtained during the independent sample t-test were not confirmed when the country variable and its interaction with the time variable were introduced into mixed effects growth models (see Table 4). Consequently, the trajectory of the development of information-processing speed during the period of primary school years does not depend on educational conditions, particularly on the quality of education at school. In both analyzed samples, information-processing speed is characterized by a nonlinear trajectory with intensive growth until the second year of schooling and a further consistent significant increase until the end of primary school education. These data are consistent with studies reporting no cross-cultural differences in the rate of development of information-processing speed during school years (e.g., [27]).

This study showed significant differences between individuals in each year of primary schooling, both in the level of processing speed and in the rate of change in processing speed. Supporting this is the fact that the model with the random slope time variable fit the data significantly better than the fixed slope model (see Table 3). These results may indicate the existence of hidden classes in the development of information-processing speed during the four years of primary school education.

Latent class growth modeling revealed that information-processing-speed development can be described by three latent classes. The most prevalent of the classes, which included 92% of the respondents of the combined sample, demonstrated linear reduction in reaction time from the first to the fourth grade. The next class, comprising 7% of the respondents, demonstrated the most intensive growth in processing speed from first to second grade, but with a further slowdown that remains statistically significant until the end of primary schooling. Finally, the last class, which includes only 1% of the respondents, shows a processing speed increase only from the first to the second grade.

It should be noted that the differences between the latent classes, while statistically identifiable, are minimal (see Table 6). In addition, one of the classes includes the vast majority of respondents—92%. These facts give grounds to consider the sample as fairly homogeneous in terms of the processing-speed development rate during primary school education. Thus, the development of information-processing speed may be mostly characterized by a consistent linear increase in processing speed across primary school. Nevertheless, the identification of hidden classes, which differ from the general trend, emphasizes the need to organize preventive measures to support some groups of schoolchildren.

The analysis of the percentage distribution of the latent classes of information-processing-speed development of Russian and Kyrgyz schoolchildren showed no significant differences. Thus, 92% of respondents corresponded to the class with linear growth in processing speed both in the Russian and Kyrgyz samples. The class with an slowdown in the growth rate of processing speed after the second grade included 6% of the Russian sample and 8% of the Kyrgyz sample. The class with no growth after the second year of study includes 2% of respondents in the Russian sample and no respondents in the Kyrgyz sample.

These results confirm the homogeneity of the analyzed cross-cultural samples regarding the trajectory of the development of processing speed during the whole period of primary school education. Consequently, the development of processing speed at the initial stage of education does not depend on the sociocultural conditions of education, particularly on markers of the quality of school education, such as overcrowding in classes, extreme teacher workloads, and the level of students’ learning. The absence of such an influence and, accordingly, cross-cultural similarity in indicators of processing speed, developmental trajectories, and their types can be explained by a stronger relationship of this cognitive indicator with the physiological and genetic background rather than with sociocultural factors [26,27,28]. Indeed, in other studies, processing speed is considered a biomarker of cognitive age and, in many works, of biological age in general (e.g., [11]).

Further research could involve a cross-cultural longitudinal analysis of the development of processing speed during secondary school education. In addition, the identification of early specific factors that enhance or weaken the rate of development of this educationally important cognitive indicator is necessary for taking preventive measures in the school education system.

## Figures and Tables

**Figure 1 behavsci-13-00873-f001:**
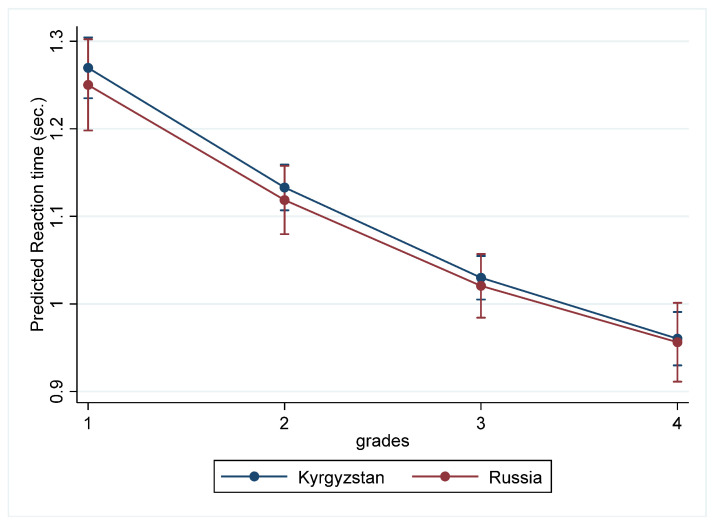
Average predicted trajectories of changes in processing speed for Russian and Kyrgyz samples.

**Figure 2 behavsci-13-00873-f002:**
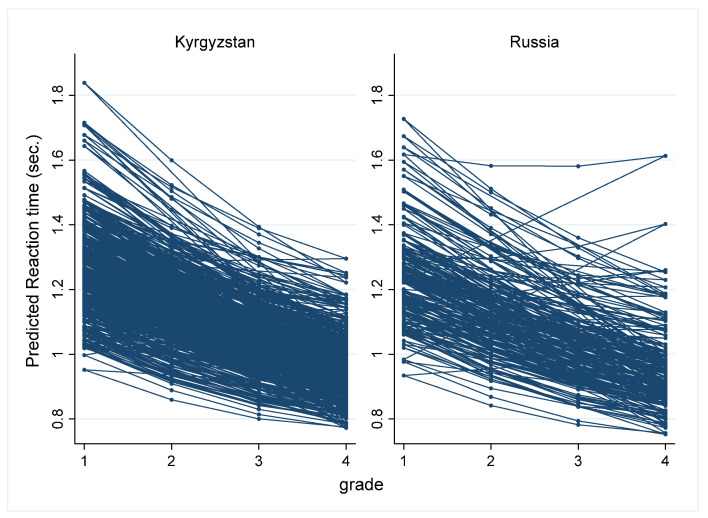
Predicted individual trajectories of changes in processing speed for Russian and Kyrgyz samples.

**Figure 3 behavsci-13-00873-f003:**
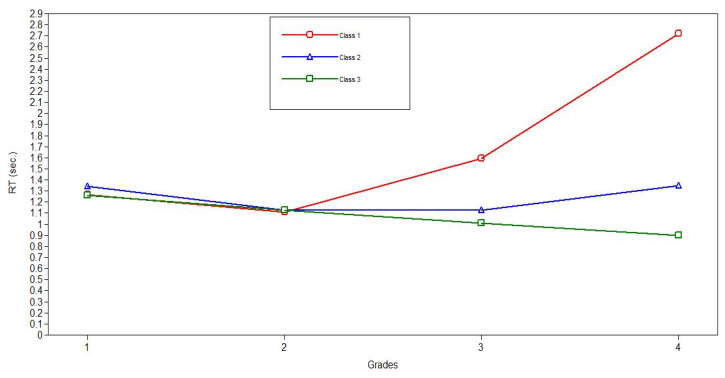
Estimated average trajectories of processing-speed developmental changes for three latent classes.

**Table 1 behavsci-13-00873-t001:** Descriptive statistics for Russian and Kyrgyz samples from Grade 1 to Grades 4.

Grade	Country	N	Mean	SD	Min	Max
Grade 1	Russia	97	1.29	0.31	0.69	2.25
	Kyrgyzstan	239	1.26	0.28	0.65	2.40
Grade 2	Russia	108	1.06	0.22	0.67	1.98
	Kyrgyzstan	272	1.14	0.27	0.70	2.47
Grade 3	Russia	114	1.05	0.39	0.60	2.83
	Kyrgyzstan	274	1.04	0.24	0.65	2.09
Grade 4	Russia	114	0.95	0.35	0.52	2.93
	Kyrgyzstan	268	0.95	0.20	0.59	1.77

**Table 2 behavsci-13-00873-t002:** Results of independent samples t-test for Russian and Kyrgyz samples from Grade 1 to Grades 4.

Variable	Russian Sample		Kyrgyz Sample		Mean Difference		*t*-Test	Cohen’s d [95% CI]
	Mean	SD	Mean	SD	Mean	s.e.		
Processing speed at Grade 1	1.29	0.31	1.26	0.28	0.11 [−0.12; 0.35]	0.03	0.94	0.11 [−0.12; 0.35]
Processing speed at Grade 2	1.06	0.22	1.14	0.27	−0.31 [−0.53; −0.08]	0.03	−2.71 **	−0.31 [−0.53; −0.08]
Processing speed at Grade 3	1.05	0.39	1.04	0.24	0.07 [−0.15; 0.29]	0.03	0.66	0.07 [−0.15; 0.29]
Processing speed at Grade 4	0.95	0.35	0.95	0.20	−0.01 [−0.10; 0.10]	0.03	0.08	−0.01 [−0.10; 0.10]

** *p* < 0.01.

**Table 3 behavsci-13-00873-t003:** Results of mixed effects growth modeling on the pooled cross-country sample.

	Baseline Model	Model 1	Model 2	Model 3
Fixed effect				
Constant	1.09 *** (0.01)	1.25 *** (0.01)	1.26 *** (0.01)	1.26 *** (0.02)
Time		−0.10 *** (0.01)	−0.15 *** (0.02)	−0.15 *** (0.02)
Time^2^			0.02 ** (0.01)	0.02 ** (0.01)
Random effect				
Intercept variance	0.02	0.02	0.02	0.04
Residuals	0.07	0.05	0.05	0.05
Slope variance (time)				0.003
Covariance between intercept and slope (time)				−0.007
Log-likelihood	−263.73	−117.06	−113.06	−106.62
LR test (Δdf)		293.36 *** (1)	8.00 ** (1)	12.88 ** (2)

*** *p* < 0.001, ** *p* < 0.01.

**Table 4 behavsci-13-00873-t004:** Results of mixed effects growth modeling on the pooled cross-country sample (Models 4 and 5).

	Model 4	Model 5
Fixed effect		
Constant	1.27 *** (0.02)	1.27 *** (0.02)
Time	−0.15 *** (0.02)	−0.15 *** (0.02)
Time^2^	0.02 ** (0.01)	0.02 ** (0.01)
Country	−0.01 (0.02)	−0.02 (0.03)
Interaction effect		
Time × country		0.01 (0.01)
Random effect		
Intercept variance	0.04 (0.01)	0.04 (0.01)
Residuals	0.05	0.05
Slope variance (time)	0.003	0.003
Covariance between Intercept and Slope	−0.007	−0.007
Log-likelihood	−106.48	−106.41
LR test (Δdf)	0.27 (1)	0.15 (1)

*** *p* < 0.001, ** *p* < 0.01.

**Table 5 behavsci-13-00873-t005:** Fit indices for models with single-class growth of processing speed.

Models	BIC Sample-Adj.	χ^2^	df	RMSEA [90% CI]	CFI	SRMR
One class linear	239.78	19.58	5	0.082 [0.046–0.121]	0.91	0.055
One class quadratic	236.53	4.71	1	0.092 [0.023–0.183]	0.98	0.025

BIC sample-adj.—Bayesian information criterion (sample adjusted); RMSEA [90% CI]—root mean square error of approximation with 90% confidence interval; CFI—comparative fit indices; SRMR–standardized root mean square residual.

**Table 6 behavsci-13-00873-t006:** Fit indices for models of growth of processing speed with two, three, and four quadratic latent classes.

Number of Classes	Variance of I, S and Q	BIC Sample-Adj.	Entropy	VLMR LR Test (Sig.)	Adj. LMR LR Test (Sig.)
2	Equal to 0	238.00	0.78	0.10	0.11
2	Equal across classes	79.27	0.99	0.000	0.000
2	Estimated separately for each class	62.84	0.74	0.004	0.005
3	Equal to 0	88.32	0.84	0.08	0.08
3	Equal across classes	30.03	0.91	0.004	0.004
3	Estimated separately for each class	31.24	0.83	0.08	0.08
4	Equal to 0	42.41	0.77	0.26	0.27
4	Equal across classes	−5.431	0.91	0.33	0.35
4	Estimated separately for each class	7.94	0.75	0.19	0.20

I—intercept, S—slope, Q—quadratic term; BIC sample-adj—Bayesian information criterion (sample-adjusted); Entropy–a summary indicator of certainty in classification; VLMR LR test—Vuong–Lo-Mendell-Rubin likelihood ratio test; Adj. LMR LR test—Lo–Mendel–Rubin likelihood ratio test.

**Table 7 behavsci-13-00873-t007:** Parameters of latent classes (model with three latent classes).

Latent Class	Intercept	Slope	Quadratic Term	Proportion of Participants	Posterior Probabilities
First class	1.27 *** (0.06)	−0.48 (0.32)	0.32 ** (0.11)	0.01	0.92
Second class	1.34 *** (0.08)	−0.33 *** (0.09)	0.11 *** (0.02)	0.07	0.89
Third class	1.26 *** (0.02)	−0.14 *** (0.02)	0.01 (0.01)	0.92	0.97

*** *p* < 0.001, ** *p* < 0.01. Proportion of participants—proportion of participants based on their most likely latent class membership; Posterior probabilities—average latent class probabilities for most likely latent class membership.

**Table 8 behavsci-13-00873-t008:** Proportion of latent classes within Russian and Kyrgyz samples.

	First Latent Class	Second Latent Class	Third Latent Class
Russia	2%	6%	92%
Kyrgyzstan	0%	8%	92%

## Data Availability

The data presented in this study are available on request from the corresponding author. The data are not publicly available due to the terms of the grant funding.

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
