# Peer review of "Processing Speed throughout Primary School Education: Evidence from a Cross-Country Longitudinal Study"

_behavsci, 2023, doi:10.3390/bs13100873_

Round 1

Reviewer 1 Report

Authors may provide more general details on the differences in the quality of schooling (p. 3) and relevant cultural differences between Russia and Kyrgyzstan, in order to reinforce the link between these variables and the results. Since the role of cross-cultural aspects is mentioned frequently, the reader is given the idea that it is extremely relevant.

Author Response

Dear Reviewer #1,

Thank you for your suggestion.

According your comment about addition of general details on the differences in the quality of schooling (p. 3) we are clarifying  the details, which can lead to a decrease in the quality of school education in Kyrgyzstan  within following text: “In particular, due to severe lack of teachers in Kyrgyzstan, a single primary school teacher works both morning and evening shifts, and the shortage of schools leads to a larger class size of 40 to 45 pupils in comparison with the 19–27 schoolchildren in Russian classes [20].”.

Reviewer 2 Report

The theoretical introduction is correctly written and is a good justification of the research. However, it is lengthy. I think the part of the introduction, where the authors describe changes in reaction time over the course of a person's life, including late adulthood, should be removed. This is not directly relevant to the study of children in grades 1, 2 and 3.

The article needs information about the children studied. It is not enough to refer to a previous publication because it is not in open access. It is difficult to expect a reader of an article to buy access to another article in search of data about a group of children.

Since the study lasted several years it is difficult to believe that the number of children remained constant. It is necessary to state what the number of subjects was in each year in each group.

In the description of the results, in the tables, wherever possible, the size of the relevant effects e.g. Cohen's d should be given.

Line 36 - unknown word the зrevention

Author Response

Dear Reviewer #2,

Thank you for all of your suggestions.

The changes made based on your suggestions have undoubtedly improved the quality of the manuscript.

1) We’ve shortened the text at the “Introduction" section regarding describing changes in reaction time over the course of a person's life, including late adulthood.

In particular, the following text was removed:

“A decrease in information processing speed with age was also shown in a more recent population-based study on adults aged 20 to 80 [12]. … In particular, processing speed, measured as the reaction time on a choice task, which required pressing of a key corresponding to a certain number appearing on the screen, gradually decreased over the entire age interval from 20 to 50 years. Meanwhile, processing speed, measured by a simple reaction time, which required pressing a key when any stimulus appeared on the screen, started to decrease only at the age of approximately 50 years [12]. In old age, the speed characteristics of information processing noticeably worsen; there is a significant slowdown in the speed of information processing, not only in comparison with young people but also in comparison with less senior elderly adults aged 55 to 87 years [15] or 65 to 94 years [16].“. Please, see p.2, paragraph 1.

2) We’re added information about the children studied to "Materials and Methods” section (2.1. Participants, p. 4).

We’re added following text:

The Kyrgyz sample included 303 participants (46% were girls). The mean age in the first grade was 7.46 years (SD = 0.39, range 6.42–8.83) and that in the fourth was 10.40 years (SD = 0.39, range 9.33–11.83). Sixty-six percent of the participants were Kyrgyz, 10% were Russian, and the rest belonged to other ethnic groups (e.g., Dungan, Uyghur, Kazakh). Some participants took part only once; thus, their data were removed from the analysis. Fifty-nine percent of the pupils participated in four waves, 31% in three waves and 10% in two waves.

The Russian sample included 138 participants (46% were girls) from one primary school. The mean age in the first grade was 7.84 years (SD = 0.34, range 7.06–8.37), and that in the fourth was 10.77 years (SD = 0.36, range 9.72–11.85). One hundred percent of the participants were Russian. Some schoolchildren participated only once; thus, their data were removed from the analysis. Forty-eight percent of the pupils participated in four waves, 38% in three waves and 14% in two waves.

3) At Table 1 (p. 6) the number of subjects was in each year in each group was added.

4) At Table 2 (p. 7) the size of the relevant effects (Cohen's d & 95% CI) was given.

5) The word “the зrevention” corrected to “the prevention” (p. 1).